# The Impact of Health Geography on Public Health Research, Policy, and Practice in Canada

**DOI:** 10.3390/ijerph20186735

**Published:** 2023-09-09

**Authors:** Michelle M. Vine, Kate Mulligan, Rachel Harris, Jennifer L. Dean

**Affiliations:** 1Department of Health Sciences, Brock University, St. Catharines, ON L2S 3A1, Canada; 2Canadian Institute for Social Prescribing, Dalla Lana School of Public Health, University of Toronto, Toronto, ON M5T 3M7, Canada; kate.mulligan@utoronto.ca; 3Independent Researcher, Hamilton, ON L8P 1H6, Canada; rach.l.harris@gmail.com; 4School of Planning, University of Waterloo, Waterloo, ON N2L 3G1, Canada; jennifer.dean@uwaterloo.ca

**Keywords:** geographies of health, public health, healthy environments, social determinants of health

## Abstract

The link between geography and health means that the places we occupy—where we are born, where we live, where we work, and where we play—have a direct impact on our health, including our experiences of health. A subdiscipline of human geography, health geography studies the relationships between our environments and the impact of factors that operate within those environments on human health. Researchers have focused on the social and physical environments, including spatial location, patterns, causes of disease and related outcomes, and health service delivery. The work of health geographers has adopted various theories and philosophies (i.e., positivism, social interactionism, structuralism) and methods to collect and analyze data (i.e., quantitative, qualitative, spatial analysis) to examine our environments and their relationship to health. The field of public health is an organized effort to promote the health of its population and prevent disease, injury, and premature death. Public health agencies and practitioners develop programs, services, and policies to promote healthy environments to support and enable health. This commentary provides an overview of the recent landscape of health geography and makes a case for how health geography is critically important to the field of public health, including examples from the field to highlight these links in practice.

## 1. Introduction

Health geography studies how human health and health systems are diffused, distributed, determined, and delivered by using a spatial lens to examine these factors across a range of scales [1,2]. Indeed, the places we occupy—where we are born, where we live, where we work, and where we play—have a direct impact on our health and our experiences of health.

The Canadian Public Health Association defines public health as “the organized effort of society to keep people healthy and prevent injury, illness and premature death” [3]. Public health focuses on the health of the whole population. This focus is different but not disconnected from other parts of the health system, which often prioritizes individual health. A population health approach aims to improve the health of the population and to reduce inequities in health among population groups by looking beyond the health-care system to consider a wider range of factors and conditions impacting health status [4]. Public health functions through programs, services, policies, and other interventions designed to improve the population’s health. These activities are undertaken by multiple levels of government and in collaboration with partners across sectors, such as community and healthcare organizations.

Health geographers have made important contributions to public health scholarship on the reciprocal relationship between place and human health. Most notably, geographers have acknowledged that contextual, compositional, and collective aspects of place together influence population health in a much more relational rather than unilateral way [5,6]. The importance of taking a multiscalar and intersectional approach has also been advocated by health geographers engaging with the WHO’s Healthy Cities initiative, which adopts a community approach to address urban health inequality [7], and more recently, the integrated One Health approach that seeks to explore planetary and human health through connections between humans, animals, and ecosystems [8]. These approaches counter the false dichotomy between population and individual determinants of health (i.e., public health vs. the healthcare system), as Macintyre et al. [6] (p. 128) conclude “… rather than there being one single, universal ‘area effect on health’ there appear to be some area effects on some health outcomes, in some population groups, and in some types of areas.”

The factors and conditions impacting health status are also known as the determinants of health. It is estimated that factors such as behaviour, biology, and access to and quality of healthcare only determine about half of an individual’s health outcomes. The other half of these outcomes are determined by the conditions in which people are born, grow, work, live, and age. These conditions are known as the social determinants of health and include factors such as income, education, working conditions, race, gender, and culture [9].

In Canada and worldwide, there are historical, persistent, and emerging health inequities. The term inequity (or health disparity) is used to: “describe those health inequities, though avoidable, are not avoided and hence are unfair” [10] (p. S517). Individual and social determinants both directly influence these health inequities. To address these inequities in health outcomes across subpopulations, health systems have a responsibility and commitment to data collection, measurement, reporting, and evaluation to highlight these inequities, with the goal of improving access to care for all individuals [11]. The difference between inequity and inequality has been described by Global Health Europe, and it is important to note, “Inequity refers to unfair, avoidable differences arising from poor governance, corruption or cultural exclusion while inequality simply refers to the uneven distribution of health or health resources as a result of genetic or other factors or the lack of resources” [12].

Income-based inequity has historically been a primary indicator of relative rank in the social hierarchy. There has been a call to expand this further to consider alternative “axes of stratification” that include inequities in human capital and political power, in addition to cultural and social assets [13]. Widening discrepancies in income inequality have been, in part, explained through various social markers, including disparities in job security, housing provision, and employment, all of which shape health [14]. These inequities are the result of occurrences resulting from ongoing power relations related to colonial legacies, the slave trade, and employment into insecure and unpleasant occupations through immigration [14]. For example, in Canada, colonial strategies, such as the Indian Act and residential schools, intended to assimilate indigenous people by severing relationships between children and families, families from their land/territory, and nations from their culture [15]. These historic and ongoing impacts of colonialism have contributed to indigenous health inequities that are systematic, socially produced, and unjust. Other types of inequities have been examined by health geographers. Research on the role of discrimination in shaping health and health inequities has focused attention on ethnicity, homelessness, disability, nomad populations, sexual orientation, and gender by way of social, cultural, physical, and economic environments [14].

Health geographers are curious about understanding the spaces and places that influence our health. Furthermore, health geography’s focus on examining the processes and relationships across time and space is valuable to public health, which always considers historical trends, emerging threats, and working at multiple scales. In this commentary, we provide an overview of the recent landscape of health geography and make a case for how health geography is critically important to the field of public health, including examples from the field to highlight these links in practice. Health geographers are well-positioned to continue to be strong collaborators in the work needed to transform Canada’s public health system.

## 2. Approaching Public Health through Health Geography

Public health is influenced by health geography in multiple ways, from the unseen assumptions and beliefs about the world and how it can be known to the specific practices and methods for gathering knowledge about the world. Unlike the field of medicine, which is rooted in objective and positivist approaches to understanding human health through scientific inquiry, health geographers have adopted various ontological, epistemological, and methodological approaches in public health practice [16]. These approaches support and allow health geographers to examine the processes and relationships across time and space that influence human health and diseases as they evolve [17]. Ontology comprises theories that seek to answer questions about the conditions of the world for knowledge to be possible. Theories are developed by examining the way in which individuals experience life or highlight differences and similarities within everyday life, both inductively (deriving general assumptions from specific instances in life) and deductively (deriving specific assumptions from general instances in life) [18]. Positivist theorists draw upon quantitative statistical methods to investigate patterns in health across space and place, with a sample large enough to generalize to the population, for example, regarding disease spread or diffusion [19,20]. Meanings of health and disease have been subjectively explored by health geographers through experiences of places that are socially constructed, known as social interactionism [21,22,23]. (Giddens emphasized structuralist theory as the practical use of society’s structural components (rules, norms, institutions), which exist as real when they are generated through the social practices of humans [22].

Epistemology is the investigation of the origin, methods, and limitations of what we know about the world—and how we know it—and how that establishes what we accept as valid knowledge [18,23]. In her commencement address to The School of Public Health at the University of California at Berkeley, Krieger [24] acknowledged her belief that public health practice and research requires a passionate epistemology, specifically, “a way of knowing that is at once critical, rigorous, humble, and partisan, on the side of all who are burdened by premature mortality, preventable disease … where knowledge is instrumental for closing, not widening, gaps between rich and poor, between the powerful and disempowered, within countries and across nations” [24] (p.288). It is within this epistemological context that health geographers are often situated.

Methodologically, health geographers adopt a coherent set of procedures and rules to investigate a phenomenon, guided by a framework identified as part of its ontology and epistemology [23] to collect data (i.e., quantitative, qualitative, spatial analysis) to examine our environments and their relationship to health. Quantitative data collection methods have been used to examine geographies of disease (both infectious and chronic), the food-obesity-built environment [25], walkability [26], green spaces [27], and access to healthcare services by way of multilevel modelling and spatial analysis [28] using geographic information systems (GIS) mapping [29,30], personal activity devices/trackers [31,32] with an emphasis on marginalized populations in particular geographical locations (women, low income, racialized populations) [33,34].

Qualitative methods have been used by health geographers to explore the relationship between the environment and health [35], including therapeutic landscapes [36], environmental health [37,38], disability studies [39], health outcomes in indigenous populations [40], newcomers [41,42], children [43], and aging populations [44,45].

Mixed methods—combining qualitative and quantitative research—have been utilized by health geographers to explore the relationship between the environment, human health and health and social services. For example, Chadwick and Collins [46] examined social support availability, neighbourhood characteristics and mental health of recent immigrants using in-depth interviews and secondary analysis of Canadian Community Health Survey data. International studies represent important examples that Canada can learn from. In a study on social isolation and loneliness, Finlay and Kobayashi [47] utilized a parallel convergent mixed-methods case study, including in-person, in-depth interview sessions with US older adults framed by the Neighbourhood Design Characteristics Checklist (NeDeCC) to assess residential environments at three levels: dwelling, street and neighbourhood, in-person observations, and ArcGIS mapping software to calculate the NeDeCC for every participant’s home location. In Australia, Brown and others [48] conducted a mixed-method participatory GIS study focused on evaluating the validity of using qualitative interviews and quantitative mapping methods in Australia. A mixed methods study assessing food environments in a low-, middle-, and high-income community in a Mexican city analyzed the density and proximity of food outlet types and the quantity, variety, pricing, promotion and quality of food (quantitative method), in addition to undertaking qualitative photo elicitation, which uses images during interviews to stimulate culturally relevant reflections [49].

The political ecology of health and disease has guided some of the work of health geographers to expand their understanding of health and disease by focusing on interactions between social institutions, political interests, and human−environment interactions [50]. Political ecologies of health and ill health have applied mixed research methods and multiscalar analysis across time and space [51]. In this way, there has been a need to understand how human health and culture shape interactions with the environment, including confronting the perpetuation of structural inequalities and power relations that shape cultural practices and health processes [52].

A political ecology approach has been applied by health geographers through the lens of understanding the social construction of health and illness and social relevance in a study of infant mortality [53], in addition to an examination of the HIV/AIDS epidemic in South Africa [51]. Political ecology informed an investigation of First Nation’s perspectives on the risks and benefits of salmon aquaculture development in British Columbia, Canada [54]. Strong links were made between poor health and reduced environmental resource access since there are restrictions on social, economic, and cultural activities for indigenous communities that result in good health and well-being [54]. Lead poisoning in North Carolina was examined to reveal the place-based conditions in which higher rates of ill health emerged by way of social, historical, economic, and political processes [55].

## 3. Public Health

### The Social Determinants of Health

The field of public health is an organized effort to promote the health of its population and prevent disease, injury, and premature death. Public health agencies and practitioners plan, develop, implement, and evaluate programs, services, and policies to promote healthy environments to support and enable health. It is within this context that we make the case that health geography has made significant contributions to the field of public health.

There is growing concern about public health issues impacting people’s health at the population level. According to the World Health Organization (WHO) health is a fundamental right, defined as “a state of complete physical, mental and social well-being and not merely the absence of disease or infirmity” [56] (p. 1). The field of public health focuses on the promotion and protection of health and includes the prevention of both chronic (e.g., diabetes, heart disease, cancer) and infectious diseases (e.g., COVID-19, *Clostridium difficile* (*C. diff*), Avian influenza (H5N1)) [56,57].

The social determinants of health (SDOH) refer to specific factors outside of the healthcare system or healthcare sector that influence the health of individuals and populations and reflect one’s location in society by way of income and social status, race, gender, education and literacy, employment status, and working conditions [9]. The SDOH vary between individuals and populations, where having higher levels of income leads to better health outcomes and, conversely, lower levels of income are associated with poorer health outcomes, which is referred to as the social gradient [58,59]. The social gradient also demonstrates the way income influences other SDOH (e.g., food, housing, education, health services, etc.) to produce better or worse health outcomes.

There have been recent calls to re-examine and continue expanding the SDOH, since they omit structural racism and the health system itself as a SDOH [60]. This call is relevant considering the emergence of the COVID-19 pandemic, where the lived experiences of racial and ethnic minority residents [61], food bank users [62], and essential workers were not considered in stay-at-home orders and social distancing recommendations [60]. This latter example demonstrates that structural racism is a root cause of racial health disparities inherent within the economic and employment system, in part by way of compensation and benefits and discrepancies in wage and worker safety laws, but particularly and perhaps most importantly, given that home healthcare workers are predominantly women of colour [58]. Similarly, in exploring why COVID-19 infections disproportionately impacted racialized communities, it was found that racialized individuals were more likely to be employed as essential workers or in other occupations with more exposure to infections, proximity to others, or less ability to work from home [63].

Structural determinants of health and the root causes of inequities have been examined by Crear-Perry and others [64]. Specifically, these are social and political structures—racism, classism, and gender oppression—that result in discriminatory policies and practices (e.g., human slavery, Jim Crow: state-led racial segregation, redlining: home mortgages being denied on the basis of race), that limited the socio, economic, and physical mobility and well-being of marginalized populations [64]. Similarly, the renewed research focus on the links between the built environment and population health has also shed light on the upstream structural determinants of health [65]. Of note, health geographers Dean & Elliott [66] have explored the complex interactions between physical, social, political, and economic aspects of the neighbourhood environment that produced adolescent body weight, while Pritchard [67], Leger et al. [68], and Biglieri and Dean [31] have all examined how ageist ableist assumptions and practices that have resulted in the exclusion of older adults and persons living with disabilities and diverse body sizes and shapes from everyday built environments.

Significant attention has been paid by health geographers to environmental injustice. This includes the actions and policies that place groups or communities, regardless of social position, at greater health risk through increased proximity and exposure to hazardous environmental conditions or natural disasters (https://ncceh.ca/resources/blog/renewed-attention-environmental-equity-and-justice (accessed on 12 June 2023)). Health geographers have promoted environmental justice initiatives and approaches and advocated for the inclusion of community voices to better inform the development of environmental health public policy, multiscalar analysis, and interdisciplinary partnerships [69]. Future research in this area in Canada to explore health prevention and racialization is needed [70].

Structure has also been examined as a fundamental cause of health inequity in the field of public health, as a complex layer of interrelated and interacting causes embodied by the individual [64,71]. In their example of smoking behaviours and structure, Crammond and Carey [71] outline an expanded theory of structural factors that make up a habitus of smoking, including pleasure, peers, public health messaging, perceptions of risk, cultural capital, family, coolness, economic capital, aesthetics, and gender (p. 11). In the case of smoking—as well as other “marketable health hazards” such as poor-quality food and beverages and alcohol, cannabis, and other drugs—the commercial determinants of health also impact whether someone smokes [72].

The commercial determinants of health are defined as “strategies and approaches used by the private sector to promote products and choices that are detrimental to health” [73] (p. e895). Marketing, lobbying, corporate social responsibility strategies, and extensive supply chains are channels that are driven by the internationalization of trade and capital, the demand for growth, and the expanding outreach of corporations [73]. These channels impact corporate reach and boost the health impacts related to corporate enterprise, specifically in the areas of alcohol, sugar-sweetened beverages, and tobacco sales.

There are some notable challenges associated with measuring the impact of social, structural, and behavioural determinants of health on populations, including that they are complex, multifaceted pathways with factors that are not linear and may be impacted by variables such as epigenetics and genetic factors, that health effects manifest over long periods and are hard to track, and that it is difficult to access information across sectors (e.g., education, health services, planning, housing) [74]. Further, when SDOH data are collected, they are not always connected to other data systems at the individual or patient level or available in real-time [75]. Aligned with Goal 3 of the United Nations’ Sustainable Development Goals (SDG), “To ensure healthy lives and promote well-being for all at all ages” [76], public health practitioners, researchers, and policymakers seek to improve population health and reduce differences in health outcomes (health inequities) that result from race/ethnicity, geographic location, and socioeconomic position, as identified by education, income, and wealth [77]. Without adequate data to measure, understand, and respond to health inequities through focused interventions, this becomes even more challenging.

## 4. Health Geography’s Contributions to Public Health

In the 1990s, a shift from medical to health geography included a recognition of the need to explore health and illness in the context of experiences of being in place, defined as the local environment in which social process, health, and disease occur [78,79]. A focus on place involves understanding the relationship between individuals and the physical, cultural, and social environments in which they are located, which have a direct influence on their health and, to a lesser extent, also influence their access to health services [78].

Along with this disciplinary shift, health geographers expanded their focus beyond largely positivist (objective, quantitative) approaches to those informed by theory and more qualitative in nature [80,81]. As such, by the end of the twentieth century, health geographers were engaged in research adopting a population health perspective, particularly given their renewed definition of health (as described above) and a need to understand the social determinants of health [80]. Specifically, the way in which the environment and its conditions operate to influence and/or shape people’s health has been the focus of the work of health geographers. For example, access to health-promoting or protecting resources varies between social and physical environments, resulting in patterns that shape health-related behaviours within certain population groups and thus result in health inequities [82]. In 2009, Luginaah raised an important question about where the subdiscipline of health geography was headed, specifically, “How can we continue to identify, classify and reduce the risks of health that result from environmental and social inequalities, behavioural determinants (without victim blaming) and often location-specific determinants?” [83] (p. 94). Luginaah also stated the need for health geographers to highlight the policy implications of research findings and the relevance of research to the public (health) agenda [83].

A shift towards focused efforts on public health initiatives in health geography has identified health and place as critically important for engagement between health geography and public health [2]. Health and place (housing, neighbourhood, location, infrastructure) have been expressed through the link between its quality (housing, neighbourhood deprivation) and relationship to health behaviours (physical activity, nutrition, smoking, and alcohol consumption) to inform the development of new public health interventions, urban planning initiatives, and community development [20]. Specifically in practice, health geographers have used population health surveillance by collecting and analyzing health data, health promotion by empowering individuals and communities, and injury and illness prevention through risk reduction efforts, in turn, developing health promotion and protection interventions [20]. In addition to contributing to disciplinary discussions and debates in geography, the work of health geographers continues to inform the development of recommendations for public health practice and policy [84,85].

An overarching goal of health promotion and disease prevention is to support, and not hinder, healthy behaviour. A focus on the environment, including un/supportive environmental factors, has included research on the relationship between individual behaviours and features of the built, social, and policy environments at the local level to examine immigrant well-being [41,86], child and older adult mobility [32], and overall walkability [26,31,68,87], food retail stores and food deprivation [62,88], urban agriculture [89], school nutrition policy [90,91], therapeutic landscapes (coasts, seaside, urban parks, hospitals and clinics, gardens, etc.) [92,93], dengue fever and governance [94], and overweight and obesity [95,96,97,98]. Implications of many of these studies include multicomponent public health interventions to implement within these environments in order to support health behaviours (e.g., physical activity, walkability, nutritious food purchasing and intake) or advocate for broader structural changes.

Inequities in health were examined during the COVID-19 pandemic, linking inequities in COVID-19 outcomes with existing inequities in the social determinants of health and chronic diseases [99,100]. A recent study tracked the geography of disparity connected to COVID-19 vulnerability and social determinants of health in Colorado using geospatial statistics and GIS to estimate census tract level rates of COVID-19 to map areas of low and high incidence, including links to mental health and chronic conditions and the following social determinants of health, which represent inequities in income, education, access to healthcare, and race/ethnicity [101]. In Ontario, local public health units also use GIS to estimate census tract level rates for both incidence of COVID-19 and COVID-19 vaccine uptake (City of Toronto: https://www.toronto.ca/community-people/health-wellness-care/health-programs-advice/respiratory-viruses/covid-19/covid-19-pandemic-data/covid-19-vaccine-data/ (accessed on 3 June 2023; City of Hamilton: https://www.hamilton.ca/people-programs/public-health/diseases-conditions/coronavirus-covid/covid-19-vaccine#vaccine-distribution (accessed on 3 June 2023); City of Hamilton (cases): https://www.hamilton.ca/people-programs/public-health/diseases-conditions/coronavirus-covid/covid-19-data#incidence-rate-by-ct (accessed on 3 June 2023). These data were used to inform vaccine clinic planning as well as neighbourhood-level vaccine uptake strategies. The City of Toronto, Canada collects data on neighbourhoods to address planning needs and the social determinants of health. As such, Toronto developed a neighbourhood map (N = 158) to track the weekly dose count of the population vaccinated for COVID-19 during specific periods according to neighbourhood of residence. As such,

Health inequity can explain why some neighbourhoods may have lower vaccination rates. Health inequity refers to preventable differences in health between groups of people … Inequities in vaccination uptake are often a result of lack of availability of services in one’s neighbourhood, lack of flexible service hours and access to transportation to attend vaccine clinics… In addition, lower vaccine uptake has been linked to negative historical experiences with health care institutions and distrust of health care, much of which stems from discrimination, systemic racism, and the effects of colonization[102] (p. 5).

### Population Health Intervention Research

Population Health Intervention Research (PHIR) seeks to develop and evaluate interventions (policies and programs) that influence health at the population health level while considering health equity and the contexts in which interventions are designed, established, implemented, and evaluated [103]. These include deliberate efforts to improve health across various sectors: health, education, employment, or housing, for example [104]. COVID-19 vaccination campaigns, taxation on tobacco products and sugar-sweetened beverages, school board nutrition policy [90,91], social prescribing [105,106], and infrastructure for active transportation include examples of interventions that can positively affect upstream determinants of health (e.g., risk exposure, social inequities, disadvantage) that have been explored by health geographers. These types of efforts support the impact on population health, and with a particular focus on place and context, also operate to disrupt the environmental and social conditions of risk and reduce inequities in health across settings [104]. In this way, health geographers are well positioned to undertake PHIR by way of innovative methods and theories with the goal of being able to articulate the complexity of relationships between health, interventions, and place [107].

Topics of interest to health geographers in PHIR have included housing as an important setting for improving health, such as in the contexts of improvements to safety, tenant interactions with a house, material conditions and mental health, and the symbolism and meaning of being housed after homelessness [108,109]. Another example of PHIR is using population health data via data linkage and GIS spatial analysis to explore the relationship between health outcomes and health behaviours. For example, Public Health Ontario produces interactive map-based dashboards to highlight temporal and spatial trends of key public health indicators (e.g., chronic diseases, injuries, health behaviours, health equity, mortality, reproductive and child health, and substance use) examined by public health units and the province of Ontario (https://www.publichealthontario.ca/en/data-and-analysis/commonly-used-products/snapshots (accessed on 2 June 2023). Public Health Ontario and researchers at the MAP Centre for Urban Solutions at St. Michael’s Hospital (Unity Health Toronto) have also developed the Ontario Marginalization Index (ON-Marg) (https://www.publichealthontario.ca/en/Data-and-Analysis/Health-Equity/Ontario-Marginalization-Index (accessed on 15 May 2023)), which measures and maps differences in marginalization across Ontario. Data are available at multiple levels, including at dissemination areas (i.e., population between 400 to 700 people) and public health unit levels) [110].

More generally, the built (physical) environment has seen a strong focus in health geography using geocoding (e.g., postal codes, census subdivisions (municipalities), metropolitan areas, divisions, economic regions, federal electoral districts, population centres, census tracts, and dissemination areas) [111], and Ontario Ministry of Health public health unit boundaries to layer onto population-level data [112] and to highlight Canadian health service access gaps using GIS story maps to help connect communities to health services they may not be accessing [113]. Further, the deep collaboration between health geography and urban planning has also made important theoretical and practical contributions to the development of healthy and equitable communities through a range of community, policy, governance, and structural changes [113,114,115,116,117].

The Dalla Lana School of Health at the University of Toronto recently hosted a 2023 Geospatial Data Visualization Challenge, whereby teams developed ArcGIS story maps to present a public health issue of choice through maps and visualization to highlight child poverty and food insecurity, the opioid crisis, physical activity and well-being (https://resources.esri.ca/news-and-updates/the-2023-university-of-toronto-geospatial-data-visualization-challenge (accessed on 6 May 2023)). ArcGIS developed a COVID-19 Health Dashboard to produce maps illustrating the number of COVID-19 cases by province and confirmed cases by provincial public health units, and population density at the dissemination area and neighbourhood level (https://resources-covid19canada.hub.arcgis.com/apps/90fdd2da4bba4c79a33c2202760b3c5d/explore (accessed on 6 May 2023)).

## 5. Future Directions in Health Geography and Public Health

The impacts of the COVID-19 pandemic in Canada and around the world are unprecedented and include the worsening of pre-existing health inequities [61,62,99,100,102]. The 2021 Annual Report on the State of Public Health in Canada from the Chief Public Health Officer (CPHO) reflects on the impacts of COVID-19 and highlights the criticality of community participation and collaboration to respond to current and future public health challenges [118]. One of the independent reports commissioned to inform this report argued that “the future of public health is at the neighbourhood scale” [119]. Indeed, some of the most impactful public health interventions during COVID-19 were community and neighbourhood-driven, including locally organized pop-up vaccination clinics, community ambassador programs, and community organizing to prevent transmission and encourage vaccine uptake. The public health system in Canada functions at multiple scales, and there is increasingly a need to work at the smallest scale possible, given geography’s strong influence on health.

At a larger scale, health geographers are well positioned to address health system strains and health human resource challenges in the Canadian context by undertaking additional research at the interface between public health and health services; for example, into the practice of social prescribing, or community-led referral to services to address needs rooted in the social and environmental determinants of health [120]. Health geographers are also ideally situated to interrogate the ways in which digital approaches to public health, including online health literacy and artificial intelligence, and the impacts of new technologies on public health (e.g., automated and electrified transportation) have the potential to both mitigate and exacerbate geographic disparities in access to health and health infrastructure [85,121,122].

Finally, health geographers have the potential to fill a leadership role with respect to knowledge integration across scales and disciplines necessary to address the impending public health challenges. The increasing recognition of systems complexities in ongoing public health polycrises and syndemics [123], from the opioid crisis to climate change, will require health geographers to advance understanding and action on the policy and practice dimensions of individual, community, and institutional resilience.

## Data Availability

Not applicable.

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
