# Peer review of "The Impact of Health Geography on Public Health Research, Policy, and Practice in Canada"

_ijerph, 2023, doi:10.3390/ijerph20186735_

Round 1

Reviewer 1 Report

The introductory part of the paper should focus more on modern theoretical background of medical geography around the world (not only Canadian research) and its impact on public health research. In case of the chapter 2 probably it would be better to use a graphic approach to show the methodology of the research. As for the chapter 4 it worth more highlighting and structurising fields of medical geography connected with public health research, with the help of graphic approach. The last chapter 5, entitled 'Future directions in health geography and public health' corresponds only partially with the title of the paper. It needs to be rewritten, taking into account global aspect of the current studies. 

Author Response

Peer Reviewer 1

The introductory part of the paper should focus more on modern theoretical background of medical geography around the world (not only Canadian research) and its impact on public health research.

Thank you for this feedback. We have considered it, in addition to that of Reviewer 2 and 3, and decided to focus the manuscript on the Canadian context, and thus, have changed the title of our manuscript to: “The impact of health geography on public health research, policy and practice in Canada”.

In case of the chapter 2 probably it would be better to use a graphic approach to show the methodology of the research.

Thank you for this suggestion. While we respectfully disagree with the need for a graphic approach to this section, we have added a brief introductory section to enhance the clarity and flow of the contents.

As for the chapter 4 it worth more highlighting and structurising fields of medical geography connected with public health research, with the help of graphic approach.

We respectfully disagree with the necessity of visualizing this section graphically. The section has been carefully structured to highlight the connections between the field of health geography and public health research through text.

The last chapter 5, entitled 'Future directions in health geography and public health' corresponds only partially with the title of the paper. It needs to be rewritten, taking into account global aspect of the current studies.

Thank you for the feedback. We have enhanced this section (Lines 425 - 455) and ensured that it explicitly aligns with our title and topic, which now focuses exclusively on Canada.

Reviewer 2 Report

This commentary paper overviews the recent landscape of health geography including examples from the field to highlight these links in practice. The paper is fundamentally well written. However, this paper mainly focuses on the issues in Canada. In conclusion, the authors stated “Health geographers are well-positioned to be continue to be strong collaborators in the work needed to transform Canada’s public health system”. This would be main target in this paper. Hence, it would be better stating and highlighting the country in the title and throughout the manuscript. The title shall be “The impact of health geography on public health research, policy and practice in Canada”. Further, it is unclear the significant objectives and aims in this commentary paper. It may clearly state them understanding the importance and significance publishing the paper as well.

Author Response

Peer Reviewer 2

This commentary paper overviews the recent landscape of health geography including examples from the field to highlight these links in practice. The paper is fundamentally well written. However, this paper mainly focuses on the issues in Canada.

In conclusion, the authors stated “Health geographers are well-positioned to be continue to be strong collaborators in the work needed to transform Canada’s public health system”. This would be main target in this paper. Hence, it would be better stating and highlighting the country in the title and throughout the manuscript. The title shall be “The impact of health geography on public health research, policy and practice in Canada”.

Thank you, we appreciate this feedback and have decided to focus the manuscript on the Canadian context, and thus, have changed the title of our manuscript to: “The impact of health geography on public health research, policy and practice in Canada”.

Further, it is unclear the significant objectives and aims in this commentary paper. It may clearly state them understanding the importance and significance publishing the paper as well.

Thank you for this feedback. We have added in the point you made about clarifying our objectives and purpose for writing this commentary:

In this commentary, we provide an overview of the recent landscape of health geography and make the case for how health geography is critically important to the field of public health, including examples from the field to highlight these links in practice. Health geographers are well-positioned to be continue to be strong collaborators in the work needed to transform Canada’s public health system.” Lines 102 - 107

Reviewer 3 Report

This article (commentary)  watches the topic through Canadian glasses. (This fact should be included in  the title.) We in the Central-Eastern-European region have some other determinants originated from our history. My main problem with the manuscript is the use of two  termini: inequality and inequity. As far as I know, inequity  is a social entity, the inequality is a biological one. These  priciples are mixed in this ms. For example (line65): race is a biological determinant (not social), gender is OK, but sex would be also biological.

Another note to line 188.: at the first mentioning the whole taxonomic name of an organism (Clostridium difficile) should be written, later it can be abbreviated. (That is why medical students do not know what does E. mean in the name of E. coli.)

In my opinion, authors should have mentioned the theory of Environmental (in)justice, as well.

Author Response

Peer Reviewer 3

This article (commentary) watches the topic through Canadian glasses. (This fact should be included in the title.) We in the Central-Eastern-European region have some other determinants originated from our history.

Thank you for this comment, which we have considered, in addition to that of Reviewer 1 and 2, and decided to be clear about focusing the manuscript on the Canadian context, and thus, have changed the title of our manuscript to: “The impact of health geography on public health research, policy and practice in Canada”.

My main problem with the manuscript is the use of two termini: inequality and inequity. As far as I know, inequity is a social entity, the inequality is a biological one. These priciples are mixed in this ms. For example (line65): race is a biological determinant (not social), gender is OK, but sex would be also biological.

Thank you for this thoughtful comment. We have given this some consideration and done some reading that we think will guide our approach. Here is what we have added in lines 75 - 81:

“The difference between inequity and inequality have been described by Global Health Europe and are important to note, “Inequity refers to unfair, avoidable differences arising from poor governance, corruption or cultural exclusion while inequality simply refers to the uneven distribution of health or health resources as a result of genetic or other factors or the lack of resources” (https://globalhealtheurope.org/values/inequity-and-inequality-in-health/).

Another note to line 188.: at the first mentioning the whole taxonomic name of an organism (Clostridium difficile) should be written, later it can be abbreviated. (That is why medical students do not know what does E. mean in the name of E. coli.)

Thank you for this comment. We have written out Clostridium difficile (C. difficile) on lines 211-212.

In my opinion, authors should have mentioned the theory of Environmental (in)justice, as well.

Thank you. We have added a small section on environmental justice, as follows:

Environmental justice includes the actions and activism needed to highlight environmental inequities that place groups or communities at a disadvantage, regardless of social position, as a result of hazardous environmental exposures or natural disasters (https://ncceh.ca/resources/blog/renewed-attention-environmental-equity-and-justice). Health geographers have examined environmental justice initiatives and approaches, acknowledging gaps in the inclusion of community voice to inform the development of environmental health public policy, multi-scalar analysis, and interdisciplinary partnerships (Masuda, Poland & Baxter, 2010). Future research in this area in Canada to explore health prevention and racialization is needed (Giang, Boyd, Ono & McIlroy-Young, 2022).  (Lines 254 - 259)

Round 2

Reviewer 1 Report

Thank you for your answers and improving the final version of the  manuscript.